# Functional Study of cAMP-Dependent Protein Kinase A in *Penicillium oxalicum*

**DOI:** 10.3390/jof9121203

**Published:** 2023-12-16

**Authors:** Qiuyan Sun, Gen Xu, Xiaobei Li, Shuai Li, Zhilei Jia, Mengdi Yan, Wenchao Chen, Zhimin Shi, Zhonghai Li, Mei Chen

**Affiliations:** State Key Laboratory of Biobased Material and Green Papermaking, School of Bioengineering, Shandong Provincial Key Laboratory of Microbial Engineering, Qilu University of Technology, Shandong Academy of Sciences, Jinan 250353, China; sqy10072@163.com (Q.S.); xugen2019@163.com (G.X.); lixiaobei722@163.com (X.L.); 17853425488@163.com (S.L.); jiazhilei19971221@163.com (Z.J.); yanyan170621@163.com (M.Y.); ccc19862127886@163.com (W.C.); aimin142857@163.com (Z.S.)

**Keywords:** protein kinase A, cellulase, signaling pathway, *Penicillium oxalicum*

## Abstract

Signaling pathways play a crucial role in regulating cellulase production. The pathway mediated by signaling proteins plays a crucial role in understanding how cellulase expression is regulated. In this study, using affinity purification of ClrB, we have identified sixteen proteins that potentially interact with ClrB. One of the proteins, the catalytic subunit of cAMP-dependent protein kinase A (*Po*PKA-C), is an important component of the cAMP/PKA signaling pathway. Knocking out *PoPKA-C* resulted in significant decreases in the growth, glucose utilization, and cellulose hydrolysis ability of the mutant strain. Furthermore, the cellulase activity and gene transcription levels were significantly reduced in the Δ*PoPKA*-*C* mutant, while the expression activity of CreA, a transcriptional regulator of carbon metabolism repression, was notably increased. Additionally, deletion of *PoPKA-C* also led to earlier timing of conidia production. The expression levels of key transcription factor genes *stuA* and *brlA*, which are involved in the production of the conidia, showed significant enhancement in the Δ*PoPKA-C* mutant. These findings highlight the involvement of *Po*PKA-C in mycelial development, conidiation, and the regulation of cellulase expression. The functional analysis of *Po*PKA-C provides insights into the mechanism of the cAMP/PKA signaling pathway in cellulase expression in filamentous fungi and has significant implications for the development of high-yielding cellulase strains.

## 1. Introduction

The regulation of lignocellulase expression in filamentous fungi involves a complex interplay of transcription factors and functional proteins [1,2]. Cell signal transduction pathways are crucial in the induction and control of filamentous fungal enzymes, particularly in environments rich in lignocellulosic substrates [3]. Among these pathways, the protein kinase A (PKA) signal pathway has been identified as a significant component in the regulatory network of cellulase expression in filamentous fungi [4,5,6]. *P. oxalicum*, an important filamentous fungus known for its lignocellulase production, provides a valuable model for studying the mechanisms of signal transduction [7,8]. Understanding these mechanisms can inform efforts to enhance the regulation network for cellulase expression in *P. oxalicum*. Key transcription factors involved in the regulation of cellulase expression in *P. oxalicum* include ClrB and XlnR, which act as activators, as well as the negative transcription factor CreA [1]. These transcription factors exhibit interactions and the synergistic regulation of cellulase expression. Structural analyses of protein complexes centered around cellulase transcription factors and functional studies of transcription-factor-associated proteins are critical in understanding and controlling lignocellulase expression.

The cAMP/PKA signaling pathway regulates cellular carbon source metabolism by controlling the expression and activity of enzymes [4,6,9]. This pathway involves the phosphorylation of target proteins by PKA, which alters their functional activity. The inactive PKA consists of two catalytic and two regulatory subunits forming a heterodimer. The binding of the regulatory subunit with cAMP releases its catalytic subunit. cAMP acts as a secondary messenger and binds to the regulatory subunit of PKA [10,11,12,13]. This binding event activates PKA, leading to the release of its catalytic subunit. The catalytic subunit then promotes metabolic processes in the cell, influencing how carbon sources are utilized [12,14,15,16]. In *Saccharomyces cerevisiae*, the cAMP/PKA signaling pathway is known to determine cell morphology [17]. However, in *Magnaporthe grisea* and *Colletotrichum trifolii*, the catalytic subunit of PKA can negatively affect the ability of these fungi to invade plant cells [18,19,20]. The cAMP/PKA pathway also plays a significant role in regulating enzyme expression. For example, in *Chaetomium globosum*, the knockout of the Gα coding gene Gna1 leads to impaired cellulase expression [21]. Similarly, in *Trichoderma reesei*, the adenylate cyclase and PKA catalytic subunit 1 are important factors that affect the vegetative growth of mycelium and the expression of cellulase genes [22].

Overall, the PKA signaling pathway is involved in regulating various essential cellular processes, including cell growth, development, metabolism, and stress response [12,14,23]. Understanding the signaling mechanisms involved in PKA regulation can contribute to the improvement of lignocellulase expression and the development of more efficient approaches for utilizing lignocellulosic biomass.

The role of PKA in the regulation of cellulase enzymes in *P. oxalicum* is not well understood. In this research, we identified the catalytic subunit of PKA, *Po*PKA-C (PDE_03213). We investigated the impact of *Po*PKA-C on the growth and development of *P. oxalicum* 114-2, as well as its effect on cellulase expression. These findings provided valuable insights into the regulatory network of filamentous fungal enzymes and have implications for the development of high-yield cellulase strains.

## 2. Materials and Methods

### 2.1. Strains and Culture Conditions

The *P. oxalicum* wild type strain (114-2, CGMCC 5302) was maintained in our laboratory. To obtain spores, the strains were cultivated on potato dextrose agar (PDA) medium at 30 °C for 96 h. The spores were harvested using 0.9% physiological saline solution. For inoculation, 1.0 × 10^8^ conidia were added to MMS (minimal medium salt solution, KH_2_PO_4_ 3 g/L, NH_4_HSO_4_ 2 g/L, MgSO_4_∙7H_2_O 0.5 g/L, CaCl_2_ 0.5 g/L, FeSO_4_∙7H_2_O 7.5 mg/L, MnSO4∙H_2_O 2.5 mg/L, ZnSO4∙7H_2_O 3.6 mg/L CoCl_2_∙6H_2_O 3.7 mg/L, 0.025 mg/L CuSO_4_∙5H_2_O/L) medium containing 2% glucose and incubated at 30 °C and 200 rpm for 24 h [24]. The mycelium was then collected through vacuum filtration, and 0.5 g of mycelium was transferred to 50 mL of cellulase-induced fermentation medium (consisting of MMS salt solution, 1% bran, and 1% microcrystalline cellulose). The cultures were incubated at 30 °C and 200 rpm for 6 days.

### 2.2. Construction of Strains

*P. oxalicum* 114-2 was modified to include *flag-ha* tags on the *clrB* gene. To identify proteins that interacted with ClrB, a ClrB overexpression strain (P*bgl2-clrB-flag-ha-ptrA*) was developed using *P. oxalicum* 3-15 as the starting strain and incorporating the strong inducible promoter *bgl2* to enhance ClrB expression. The P*bgl2-clrB-flag-ha-ptrA* overexpression construct contained the *bgl2* promoter, the ClrB-FLAG-HA coding region and terminator region, as well as the *ptrA* resistance genes. The *bgl2* promoter was amplified from *P. oxalicum* 114-2 genomic DNA using Pbgl2-F and Pbgl2-R primers. The ClrB-FLAG-HA coding region and terminator region were amplified from *P. oxalicum* 3-15 genomic DNA using (P*bgl2*) ClrB-F and (PtrA) HAClrB-R primers. The *ptrA* screening marker genes were amplified from plasmid pME2892 using ptrA-F and ptrA-R primers. The P*bgl2* promoter region, ClrB-FLAG-HA coding region and terminator region, and *ptrA* screening marker were fused together using the double-joint method [25]. The resulting P*bgl2-clrB-flag-ha-ptrA* expression cassette was amplified using primers P*bgl2*(IN)-F and ptrA-R, using the fusion product of the three fragments as the template. The P*bgl2-clrB-flag-ha-ptrA* expression construct was then transferred into *P. oxalicum* 3-15 to generate the P*bgl2-clrB-flag-ha-ptrA* overexpressed strain (*PoI-clrB*). All primers were listed in Appendix A.

The Δ*PoPKA*-C mutant strain was constructed using the homologous recombination double-exchange method. To construct the *PoPKA-C* gene knockout cassette, we amplified the upstream and downstream homology fragments of the *PoPKA-C* gene using primers 3213-F1/3213HPH-R and 3213HPH-F/3213-R1 and *P. oxalicum* 114-2 genomic DNA as a template. We also amplified the hygromycin resistance gene *hph* using primers Hph-F/Hph-R and plasmid pSilent-1 as a template. These fragments were fused together using double-joint PCR, and the resulting deletion cassette (Δ*PoPKA-C*::*hph*) was amplified using primers 3213-F2/3213-R2. The *PoPKA-C* deletion cassette was then transferred into the *P. oxalicum* 114-2 strain using a polyethylene glycol-mediated method [26] to obtain the *PoPKA-C* gene knock-out strain Δ*PoPKA-C*. Transformants were obtained and purified by two rounds of monosporation on plates containing hygromycin resistance.

The *PoPKA-C* expression cassette was amplified from the genomic DNA of *P. oxalicum* 114-2 with primers 3213-F1/(PtrA)3213-R. We also amplified the *ptrA* gene from plasmid pME2892 with primers ptrA-F/ptrA-R. The fusion of the *Po*PKA-C expression cassette and *ptrA* screening marker was achieved using primers 3213-F2 and ptrA-R. The resulting construct, *CPoPKA-C*, was transferred into the Δ*PoPKA-C* strain to create the complementation strain *CPoPKA-C*. The *CPoPKA-C* transformants were verified using primer pair 3213-F/PtrA-R. The deletion and complementation of *PoPKA-C* were further confirmed using Southern blot analysis. A 631-bp hybridization probe DNA fragment was amplified using primers 3213-F2/3213tz-R and the genomic DNA of *P. oxalicum* 114-2 as a template. The genomic DNA of *P. oxalicum* 114-2, Δ*PoPKA-C,* and *CPoPKA-C* were double digested with *Bam*HI and *Kas*pAI, respectively. The Southern blot analysis was performed using DIG High Prime DNA labelling and Detection Starter Kit I (Roche, Basle, Switzerland) according to the instruction manual.

### 2.3. FLAG–HA Tandem Affinity Purification

A total of 0.5 g of mycelium from the *PoI-clrB* strain was collected and transferred to the enzyme production medium containing 1% bran and 1% cellulose. The cultures were incubated at 30 °C and 200 rpm for 72 h. The mycelium was filtered and washed with Harvest Solution (NaCl 9.6 g/L, Dimethyl sulfoxide 10 mL/L, 100 mM Phenylmethanesulfonyl fluoride 10 mL/L). It was then ground with liquid nitrogen and added into a 50 mL centrifuge tube containing 5 mL of B250 (NaCl 14.6 g/L,1 M Tris-HCl (pH 7.5) 100 mL/L, glycerol 100 mL/L, 500 mM Ethylenediaminetetraacetic acid (pH 8.0) 2 mL/L, Nonidet P 40 1 mL/L). The mixture was shaken and mixed well, followed by a 10 min ice bath and centrifugation at 10,000 rpm at 4 °C for 30 min. The aqueous phase in the intermediate layer was carefully collected into a new centrifuge tube. (1) ANTI-FLAG M2 (Sigma, Darmstadt, Germany) affinity resin-binding target proteins: To bind target proteins to the ANTI-FLAG M2 affinity resin, the resin was first washed with centrifugation three times. Then, mouse IgG agarose (Sigma, Darmstadt, Germany) beads were added and this was incubated using a rotating shaker at 4 °C for 2.5 h. The mixture was centrifuged at 3000 rpm at 4 °C for 2 min, and the supernatant was collected in another 50 mL centrifuge tube. Three ANTI-FLAG M2 affinity resin batches were combined and incubated overnight at 4 °C on a rotary table. After centrifugation at 3000 rpm for 2 min at 4 °C, discard the supernatant was discarded and the ANTI-FLAG M2 affinity resin was transferred, bound to the proteins, into a 1.5 mL EP tube. The samples were centrifuged at 3000 rpm for 30 s at 4 °C, and the supernatant was discarded. A total of 500 μL of the cocktail was added and this was incubated at 4 °C for 10 min on a rotary shaker. The unbound protein was removed by centrifugation at 3000 rpm for 30 s at 4 °C, and the supernatant was discarded. One-step elution was performed with 3 × FLAG. (2) Target proteins were combined with ANTI-HA (Sigma, Darmstadt, Germany). After washing the anti-HA resin three times by centrifugation, 400 μL of the eluent and 200 μL of the cocktail were added, and the incubation bath was rotated at 4 °C for 120 min. Next, 500 μL of the cocktail was added and the incubation bath was rotated at 4 °C for 10 min. The centrifugation was then performed at 4 °C, 3000 rpm for 30 s to remove unbound proteins. This process was repeated once. Two-step elution was performed using 8 M urea. A total of 80 μL of 8 M urea was added and incubated at room temperature for 10 min. Next, centrifugation was performed at 4 °C, 3000 rpm for 30 s to collect the supernatant and secondary eluent. Denatured and non-denatured protein electrophoresis was performed separately on the enriched proteins, followed by silver staining and western blot verification. The eluent for mass spectrometry was also prepared.

### 2.4. Phenotype Analysis

A total of 1 μL of fresh conidia suspension with a concentration of 1 × 10^8^/mL was dotted onto agar plates containing 2% glucose, 1% cellulose, PDA, and wheat bran, respectively. The diameter of the hydrolysis halo was measured after incubating at 30 °C for 72 h. The spore production capacities of strains were measured on PDA medium. The fresh spore suspension with a concentration of 1 × 10^6^/mL was evenly spread on the plates in 100 μL volume. After 36 h and 48 h of incubation at 30 °C, the hyphae were observed under a 400× magnification to assess their microscopic morphology.

### 2.5. Determination of Strain Biomass

After culturing the conidia in MMS liquid medium containing 2% glucose for 24 h, the fermentation broth was vacuum filtered to obtain the mycelium. The collected mycelium (0.5 g) was then transferred into fresh 100 mL of MMS liquid medium containing 2% glucose. Samples were collected every 12 h, and the mycelium was harvested by centrifugation, dried to a constant weight at 65 °C, and weighed to determine the biomass of the strains.

### 2.6. Enzyme Activity Assay

The strains were grown in a liquid medium containing 1% wheat bran and 1% microcrystalline cellulose. We measured the activity of filter paper enzyme (FPA), β-glucosidase, cellobiohydrolase, and endoglucanase in the fermentation supernatant at different time points (72 h, 96 h, 120 h, and 144 h). We used specific substrates, such as Whatman^TM^ 1 filter paper, sodium carboxymethyl cellulose (CMC-Na) (Sigma, Darmstadt, Germany), *p*-nitrophenyl-*β*-D-cellobioside (*p*NPC) (Sigma, Darmstadt, Germany), and D-(-)-Salicin (Sigma, Darmstadt, Germany) to determine the enzyme activities [26]. The enzyme activity unit was defined as the amount of enzyme required to produce 1 μM of glucose or nitrophenyl (PNP) per minute under the specified conditions.

### 2.7. RT-qPCR Analysis

A total of 0.5 g of cultured mycelium was weighed and transferred to 100 mL of induction medium for enzyme production. The mycelium was collected at 4 h and 24 h after transfer, respectively, and total RNA was extracted. Following the instructions of the PrimeScript^TM^ RT and TB Green^®^ PreMix Ex Taq^TM^ II kit (Tli RNaseH Plus) (Takara, Tokyo, Japan), cDNA was synthesized through reverse transcription of the RNA. Subsequently, a RT-qPCR reaction was performed using the LightCycler^®^ 480 System (Roche, Basel, Switzerland). The operation conditions for the RT-qPCR were as follows: initial denaturation at 95 °C for 2 min, followed by denaturation at 95 °C for 10 s, annealing at 61 °C for 30 s, and 40 cycles of amplification. All reactions were performed in three biological triplicates. Appendix A lists the primers used in RT-qPCR.

### 2.8. Transcriptome Analysis

The conidiospores were cultured in a liquid MMS medium supplemented with 2% glucose for 24 h. The mycelium was then harvested by vacuum filtration and transferred to a carbon source-free medium at a concentration of 1 g per 50 mL. After a 4-h starvation culture, the mycelium was again harvested by vacuum filtration and 0.5 g was utilized for further experiments. This mycelium was then transferred to a 50 mL MMS liquid medium containing 1% microcrystalline cellulose and cultured at 30 °C with shaking at 200 rpm for 4 h. Total RNA was extracted from the mycelium and gene expression profiling was conducted using the Illumina NovaSeq 6000 platform (Illumina, CA, USA) provided by Novogene (Tianjin, China). The raw RNA-Seq data was deposited in the National Center for Biotechnology Information (NCBI) Sequence Read Archive (SRA) under the sequence reference number PRJNA1000278. Differential gene expression analysis was performed using a significance threshold of a *p*-value < 0.05 and a fold change (FC) ≥ 2. The clusterProfiler R package (v.3.0.3; R Foundation for Statistical Computing, Vienna, Austria) was used for Gene Ontology (GO) and Kyoto Encyclopedia of Genes and Genomes (KEGG) pathway enrichment analysis of the differentially expressed genes [27].

## 3. Results

### 3.1. Extraction and Identification of ClrB Complex Proteins

The *clrB-flag-ha* expression cassette was transferred into the wild strain *P. oxalicum* 114-2, and the *flag*-*ha* tag was added to the *clrB* gene at the specific site, resulting in the construction of *P. oxalicum* 3-15. The constructed ClrB overexpression construct *Pbgl2-clrB-flag-ha-ptrA* was then transferred into *P. oxalicum* 3-15, generating the *PoI-clrB* strain. The verification results are shown in Appendix A. Additionally, RT-qPCR analysis confirmed that the expression of the *clrB* gene in the *PoI-clrB* strain was 34-fold higher than that in the wild strain 114-2 (Appendix A), demonstrating the successful overexpression of *clrB* using the *bgl2(p)-clrB-flag-ha* expression cassette.

The ClrB-FLAG-HA fusion protein was successfully purified from the *PoI-clrB* strain using tandem affinity purification followed by non-denaturing gel electrophoresis. The eluate containing the ClrB-complex was analyzed using silver stain and western blot, confirming the presence of ClrB-FLAG-HA (Appendix A). Various protein types were identified by LC-MS/MS in the ClrB-complex eluate, including ClrB, RNA-binding proteins, elongation factors, and cAMP-dependent protein kinase A. These findings suggest that these proteins may interact with ClrB, highlighting the role of ClrB as a key activator of cellulase production in *P. oxalicum*. The results are summarized in Table 1, providing valuable insights into the protein interactions involved in cellulase regulation.

### 3.2. Sequence Alignment and Phylogenetic Analysis of PoPKA-C

In the tandem affinity purification of ClrB-FLAG-HA, the protein PDE_03213 was found to have a higher abundance in the eluent. It was predicted to be a 446 amino acid protein with a homology domain spanning amino acids 133–424. Phylogenetic analysis using Clustal W revealed that the amino acid sequence of PDE_03213 is closely related to PKA-C homologous proteins from *Penicillium* [28], with a sequence similarity of over 80% (Figure 1).

Comparisons with homologous proteins from other species showed that PDE_03213 shared 68%, 55.8%, and 82% sequence identity with proteins from *Aspergillus nidulans*, *Neurospora crassa*, and *T*. *reesei* QM6a, respectively. It also displayed 70% and 56% sequence similarity with the catalytic subunits TPK1 and TPK2 of *Saccharomyces cerevisiae* S288C. Based on these findings, the protein was named *P*. *oxalicum* protein kinase catalytic subunit *Po*PKA-C.

### 3.3. Construction of PoPKA-C Deletion and Complementation Strains

The cAMP/PKA signaling pathway, which involves PKA, plays a crucial role in various biological processes such as cell growth, reproduction, and cell death. To investigate the function of *Po*PKA-C, we deleted and complemented it in the *P. oxalicum* 114-2 strain through homologous recombination (Figure 2a). In order to investigate the function of *Po*PKA-C, a knockout cassette called Δ*PoPKA-C*::*hph* was designed and introduced into the wild-type *P. oxalicum* strain 114-2. Genomic DNA was extracted from the resulting transformants. PCR validation confirmed that the gene encoding *Po*PKA-C was successfully knocked out in the transformant (Figure 2b). Southern blot hybridization results demonstrated that the *PoPKA-C* deletion strain underwent a successful homologous double exchange knockout at the *PoPKA-C* locus, with no evidence of the knockout cassette integrating at other genomic locations (Figure 2c). This indicated the successful construction of the Δ*PoPKA-C*. Subsequently, a *Po*PKA-C gene complementation cassette was constructed and introduced into the Δ*PoPKA-C* deletion mutant to create the *PoPKA-C* complementation strain C*PoPKA-C*. Southern blot validation confirmed that the single-copy *PoPKA-C* complementation expression cassette was successfully integrated into the genome of the Δ*PoPKA-C* deletion mutant (Figure 2c). In addition, the deletion and complementation of *PoPKA-C* were also confirmed by RT-qPCR (Figure 2d).

### 3.4. Effect of PoPKA-C Deletion on Glucose Utilization by P. oxalicum

The growth and morphology of *P. oxalicum* strains 114-2, Δ*PoPKA-C*, and C*PoPKA-C* were assessed to examine the impact of the *PoPKA-C* deletion. The mutant strain Δ*PoPKA-C* displayed reduced hyphal growth on solid medium, particularly when glucose was the sole carbon source (Figure 3a). The colony growth rate of Δ*PoPKA-C* was noticeably slower compared with the wild-type strain 114-2. In liquid medium with glucose as the sole carbon source, Δ*PoPKA-C* exhibited a slower growth rate (Figure 3b). Furthermore, the glucose consumption rate of Δ*PoPKA-C* was significantly lower than that of strain 114-2 during 12–48 h. The C*PoPKA-C* exhibited a similar phenotype to that of the wild-type strain on glucose medium, further suggesting that the knockout of *PoPKA-C* indeed restricted the growth of the strain (Figure 3c).

### 3.5. Deletion of PoPKA-C Restricted the Expression of Cellulases

Deletion of *PoPKA-C* in *P*. *oxalicum* 114-2 resulted in significant limitations in the growth of hyphae on glucose medium and reduction in colony growth rate compared with the wild-type strain (Figure 3a). Additionally, the growth of Δ*PoPKA*-*C* on cellulose medium plates was also limited, and hydrolysis of cellulose was significantly reduced, suggesting that *Po*PKA-C might directly or indirectly affect the expression of cellulase enzymes in *P*. *oxalicum*. Enzyme activity assays revealed that Δ*PoPKA-C* had decreased filter paper enzyme activity (Figure 4a), β-glucosidase activity (Figure 4b), cellobiose hydrolase activity (Figure 4c), and endoglucanase activity (Figure 4d) compared with the wild-type strain. After 5 days of fermentation, the levels of these enzyme activities in Δ*PoPKA*-C were significantly lowered to only 75%, 53%, 56%, and 76% of the wild-type strain, respectively. The cellulase activity of the complementary strain *CPoPKA-C* recovered to 90% of the wild strain.

Overall, these findings demonstrated that *Po*PKA-C played a crucial role in cellulase expression in *P*. *oxalicum* 114-2, and its loss led to significant reduction in cellulase production.

To investigate the impact of *Po*PKA-C on the transcriptional regulation of cellulase genes in *P*. *oxalicum* 114-2, RT-qPCR analysis was conducted to assess the expression levels of key cellulase genes and regulators under cellulose induction condition. In Δ*PoPKA*-C, the expression of *cbh1* and *bgl1*, which are cellobiose hydrolase and β-glucosidase genes, respectively, were significantly reduced. After 4 h cellulose induction, the expression levels of these genes in Δ*PoPKA-C* were only 10.7% and 25% of those in 114-2, respectively (Figure 4e). After 24 h cellulose induction, the expression levels were further diminished, reaching only 31% and 43% of the starting strain’s expression levels, respectively (Figure 4f). These findings suggested that the deletion of *PoPKA-C* resulted in decreased transcriptional expression of cellulase genes in *P*. *oxalicum* 114-2.

The cellulase genes of *P. oxalicum* 114-2 are regulated by multiple transcription factors, including CreA, ClrB, and XlnR, which worked synergistically [1]. Among these factors, CreA is a key inhibitor of cellulase expression in *P. oxalicum*, and its repression effect is primarily influenced by carbon source metabolites [29,30,31]. On the other hand, ClrB is a key activator of cellulase expression. RT-qPCR analysis revealed that the expression of the *creA* gene was 3.6-fold higher in Δ*PoPKA*-C compared with the parent strain after 4 h of cellulosic induction (Figure 4g). Similarly, after 24 h of cellulosic induction, the expression of the *creA* gene in Δ*PoPKA*-C was 2.7-fold higher than in the parent strain (Figure 4h).

RT-qPCR results showed that the expression levels of cellulase-related genes in *CPoPKA-C* were restored. However, the expression levels of *clrB* and *xlnR* did not show significant changes upon *Po*PKA-C knockout. These findings suggest that the deletion of *Po*PKA-C selectively affects the activity of transcription factors involved in cellulase expression and regulates the expression of cellulase genes.

### 3.6. Effect of PoPKA-C Deletion on Conidial Production in P. oxalicum

Conidiation, the process of spore formation in filamentous fungi, is regulated by signal transduction in the cAMP/PKA signaling pathway [20,32,33]. Several studies have linked conidiation to the function of *Po*PKA-C in *P. oxalicum*. To investigate the impact of *Po*PKA-C on conidiogenesis, the conidiation of *P. oxalicum* strains 114-2, Δ*PoPKA-C*, and C*PoPKA-C* were measured on PDA plates. It was observed that Δ*PoPKA*-C exhibited a significantly earlier timing of conidiation compared with *P. oxalicum* 114-2 (Figure 5a). By 36 h of culture on solid plates, the strain Δ*PoPKA*-C already showed the formation of conidial stalks, whereas the wild strain *P. oxalicum* 114-2 still did not exhibit this feature (Figure 5a). These findings provide evidence for the involvement of *Po*PKA-C in regulating the timing of conidial production in *P. oxalicum*.

The conidiation process in filamentous fungi is regulated by various transcription factors, including BrlA, StuA, and FluC, which control the production of conidial pigment and the expression of genes involved in conidial wall synthesis. A 4-h cellulose induction resulted in a 26.3-fold, 12.7-fold, and 19.4-fold increase in the expression levels of *brlA*, *stuA*, and *fluC,* respectively, in the Δ*PoPKA*-C strain compared with the *P*. *oxalicum* 114-2 (Figure 5b). Upon a 24-h cellulose induction, the expression levels of *brlA*, *stuA,* and *fluC* in Δ*PoPKA*-C were approximately 6.3-fold, 33-fold, and 3.6-fold higher than those in *P*. *oxalicum* 114-2, respectively (Figure 5c). Our RT-qPCR analysis implies that the elimination of *Po*PKA-C dramatically escalates the expression of *brlA*, *stuA*, and *fluC* in the Δ*PoPKA*-C strain. These findings suggested that *Po*PKA-C played a crucial role in regulating the expression of genes associated with conidium formation in *P. oxalicum*, thereby influencing clonal reproduction.

### 3.7. Comparative Transcriptome Analysis

The function of the *PoPKA-C* gene was explored through a comprehensive analysis of the transcriptome of *P. oxalicum*. A comparative transcriptomic analysis was conducted on the wild-type strain 114-2 and the Δ*PoPKA-C* mutant strain, both induced by 1% cellulose for 4 h. Significant differentially expressed genes were identified based on statistical criteria (*p*-value < 0.05, fold change ≥ 2).

The deletion of *Po*PKA-C resulted in significant changes in genes expression. Among the significantly differentially expressed genes, 56.8% were up-regulated, while 43.2% were down-regulated. To further understand the impact of *Po*PKA-C deletion on *P. oxalicum* metabolic pathways, KEGG annotation and enrichment analysis were performed. This analysis revealed the enrichment of various metabolic pathways, such as biosynthesis of secondary metabolites, biosynthesis of amino acids, starch and sucrose metabolism, and valine, leucine, and isoleucine degradation. Additionally, carbon metabolism and ribosome metabolism were also enriched (Appendix A).

GO annotation analysis was conducted to gain insight into the functions of the differentially expressed genes. The results indicated that these genes were mainly involved in carbohydrate metabolism, transmembrane transport, and amino acid transport (Appendix A).

Overall, these findings suggest that *Po*PKA-C plays a crucial role in regulating gene expression and metabolic pathways in *P. oxalicum*, particularly in carbohydrate metabolism and transport processes.

The analysis of 80 genes involved in lignocellulose degradation in the annotated genome of *P. oxalicum* revealed that the expression levels of most cellulase and hemicellulase genes were significantly reduced in the Δ*PoPKA-C* (Figure 6). Specifically, the expression of the cellobiohydrolase gene *cbh1* and β-glycosidase gene *bgl1* were significantly decreased in Δ*PoPKA*-C, which is consistent with the results obtained from RT-qPCR analysis. Additionally, the transcriptional activity of genes associated with the cAMP signaling pathway, such as adenylate cyclase (PDE_08988), PKA regulatory subunit (PDE_04688), and PKA catalytic subunit 2 (PDE_09813), were minimally affected by the deletion of *Po*PKA-C. Interestingly, the expression of a low-affinity cAMP phosphodiesterase (PDE_00536) was found to be doubled in Δ*PoPKA-C*, while the transcriptional activity of a high-affinity cAMP phosphodiesterase (PDE_02983) was reduced by 80% compared with *P. oxalicum* 114-2. This suggests a close relationship between *Po*PKA-C and the transcriptional activity of cAMP phosphodiesterase.

## 4. Discussion

The cAMP/PKA signaling pathway plays a crucial role in various cellular processes in eukaryotes, such as carbon and nitrogen metabolism, and reproduction [12,14,23]. In *P. oxalicum*, the expression of cellulase is closely linked to the cAMP/PKA signaling pathway [3,4]. However, the exact mechanism is not fully understood. In this study, we identified and investigated the function of a catalytic subunit of PKA, *Po*PKA-C, in *P. oxalicum* 114-2. Our findings demonstrated that the deletion of *Po*PKA-C led to a significant reduction in the expression of cellulase in *P. oxalicum* 114-2.

Protein phosphorylation plays a vital role in the regulatory activity of transcription factors [10,11,34]. Through tandem affinity purification and mass spectrometry analysis, we identified sixteen proteins that potentially interact with ClrB (Table 1). These include ClrB, RNA binding protein, elongation factor, cAMP-dependent protein kinase A, and catalytic subunit of PKA, most of which are related to transcription and translation. We knocked out these proteins one by one and found that the expression of *P. oxalicum* cellulase was significantly reduced after knocking out *Po*PKA-C. PKA can activate or inhibit transcription factors to regulate cell functions [33,35,36]. In *S. cerevisiae*, transcription factors RAP1, Msn2, Msn4, etc., are regulated by PKA [37,38]. Msn2/4 is one of the most important downstream targets of PKA because their phosphorylation on functionally important motifs enables PKA to control the expression of a variety of genes [39]. PKA directly regulates Msn2, 4 by controlling its nuclear localization [38]. Zhu et al. showed that *Ao*PkaC1, the catalytic subunit of PKA in *Arthrobotrys oligospora* participates in a variety of biological processes by interacting with transcription factors in the nucleus, and found that StuA, a transcription factor, acts downstream of the cAMP-PKA signaling pathway to regulate conidiation and pathogenicity [33,40]. In this study, the loss of *Po*PKA-C might also affect the phosphorylation levels of ClrB, thereby influencing their regulatory functions on target genes. Moreover, the absence of PKA-C in *P. oxalicum* increased the expression of CreA. In *A. nidulans*, cAMP-dependent protein kinase A (PkaA) participates in regulating the distribution of carbon source metabolite repression regulatory protein CreA in and out of the nucleus, thereby regulating the expression activity of lignocellulose-degrading enzymes [41]. Additionally, Ribeiro et al. predicted a close correlation between PKA and the phosphorylation of cellulase regulatory protein CreA in *A. nidulans* through phosphoproteomics [36]. Therefore, the study of the regulation of cellulase by PKA-involved protein phosphorylation modification sites and protein interactions in *P. oxalicum* is crucial for unraveling the expression regulation network of cellulase.

The *Po*PKA-C protein, which was conserved in other fungi such as *T. reesei*, *N. crassa*, and *A. niger*, plays a crucial role in the growth and development of mycelia and the expression of enzymes [6,12,42]. Deleting *Po*PKA-C in *P. oxalicum* 114-2 resulted in reduced vegetative growth and impaired ability to utilize cellulose and glucose. Interestingly, the deletion also led to earlier production of conidia and increased branching ability of hyphae. Similar observations have been made in *N. crassa*, where deletion of the *Po*PKA-C homologous protein also resulted in earlier conidia production [12]. In our study, we found that the expression levels of key activators of conidial production, such as BrlA, and StuA, were significantly enhanced in the Δ*PoPKA*-C mutant. Additionally, FluC, which was an upstream regulatory activator of BrlA, was also upregulated [43,44]. This suggests a cascade of conidial regulatory genes that are influenced by *Po*PKA-C in *P. oxalicum* 114-2. It has been previously demonstrated in *A. nidulans* that StuA is one of the phosphorylation targets of PKA, further supporting the importance of *Po*PKA-C in normal conidial growth in filamentous fungi [36].

The cAMP/PKA signaling pathway is crucial for regulating cellulase expression in filamentous fungi. In *T. reesei*, both ACY1 and PKA affect the abundance of XYR1, a key cellulase activator, and are associated with light-induced cellulase gene expression [22]. In our study, we observed that the loss of *Po*PKA-C leads to a significant decrease in the expression of cellulase genes *cbh1*, *eg1*, and *bgl2* (Figure 4e,f). While there was obvious enhancement in the expression level of the carbon metabolism transcription factor CreA, the specificity of transcription regulatory factors ClrB and XlnR, which are related to wood cellulose enzyme regulation, also did not show any noticeable change (Figure 4g,h). It is possible that *Po*PKA-C affects cellulase expression through its impact on glucose utilization and metabolism.

The comparative analysis between Δ*PoPKA-C* and the original strain revealed significant differences in various metabolic pathways, including secondary product metabolism, carbohydrate metabolism, amino acid metabolism, ubiquinone and terpenoid quinone biosynthesis, and ribose-related pathways. In particular, the downregulation of 18 genes involved in valine, leucine, and isoleucine degradation was observed in Δ*PoPKA*-C (Appendix A). It is worth noting that *Po*PKA-C may influence phosphodiesterase (PDE_02983 and PDE_00536) activity, as evidenced by an 80% decrease in the expression of high-affinity cAMP phosphodiesterase (PDE_02983) and a doubling of the expression of low-affinity cAMP phosphodiesterase (PDE_00536). Phosphodiesterase is responsible for hydrolyzing intracellular second messengers (cAMP and cGMP) and degrading them to nucleoside 5-monophosphate (5′AMP or 5′GMP), thus terminating their biochemical actions [45]. cAMP plays a crucial role in regulating cell activities, and its concentration is mainly regulated by the balance between adenylyl cyclase synthesis and phosphodiesterase hydrolysis. This suggests that *Po*PKA-C could impact the efficiency of second messenger transmission in the cell signal transduction pathway [45].

## 5. Conclusions

This study focused on screening and studying the cAMP-dependent protein kinase A catalytic subunit *Po*PKA-C in *P. oxalicum*. The deletion of *Po*PKA-C resulted in a decrease in the glucose utilization rate, as well as a significant reduction in the expression activities of cellulase. This research has important implications for understanding the regulatory mechanisms of cellulose enzyme expression in filamentous fungi and the collaborative control mechanisms of other biological processes.

## Figures and Tables

**Figure 1 jof-09-01203-f001:**
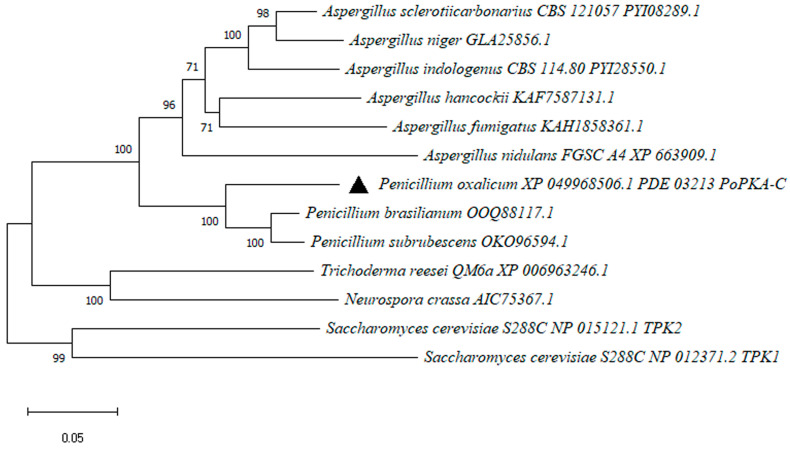
The phylogenetic analysis of *Po*PKA-C. The complete protein sequences of PKA-C homologs from *Penicillium* species, *Aspergillus* species, *T. reesei*, *N. crassa*, and *S. cerevisiae* were downloaded from the NCBI database. *Po*PKA-C is highlighted in the figure with a black triangle. The phylogenetic tree of *Po*PKA-C homolog proteins was analyzed by MEGA-X software (v.10.0.2) and the neighbor-joining method.

**Figure 2 jof-09-01203-f002:**
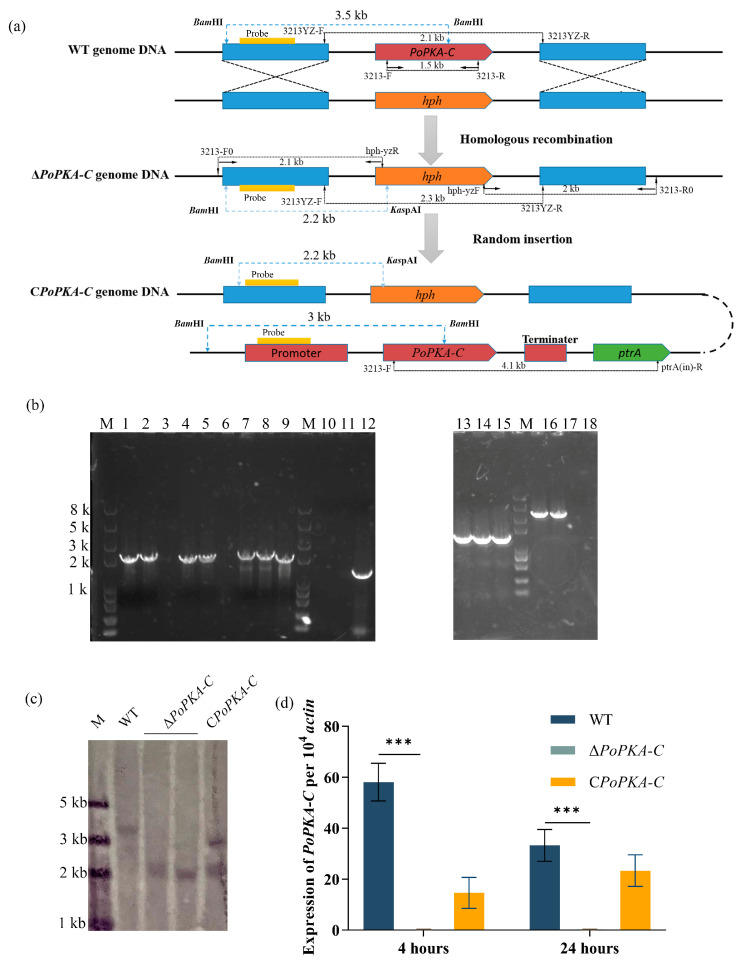
Deletion and validation strategy for *PoPK-C*. (**a**) Construction and validation strategy of Δ*PoPKA-C*. The positions of the primers, probes for Southern blot analysis, and DNA fragment sizes are marked at the corresponding positions. (**b**) PCR analysis of *PoPKA-C* gene deletion and complementation. PCR validation of *PoPKA-C* deletion was performed using primer pairs 3213-F0/hph-yzR (lanes 1–3), hph-yzF/3213-R0 (lanes 4–6), primers 3213YZ-F/R (lanes 7–9), and 3213-F/R (lanes 10–12). Complementation of *PoPKA-C* was verified using primers 3213-F/R (lanes 13–15) and 3213-F/ptrA-R (lanes 16–18). Gel lanes (3, 6, 9, 12, 15, and 18), lanes (1, 2, 4, 5, 7, 8, 10, 11), and lanes (13, 14, 16, 17) represent the results obtained from *P*. *oxalicum* 114-2, Δ*PoPKA-C*, and C*PoPKA-C*, respectively. Lane M is the Trans2K Plus II DNA marker (TransGen Biotech, Beijing, China). (**c**) Southern blot analysis of *PoPKA-C* mutant strains. The genomic DNA of *P*. *oxalicum* 114-2, Δ*PoPKA-C*, and C*PoPKA-C* were digested by *Bam*HI and *Kas*pAI double enzyme, respectively. A 621 bp fragment of the upstream homologous arm of *PoPKA-C* gene was amplified to prepare the hybridization probe. *P. oxalicum* 114-2 genomic DNA was digested with *Bam*HI enzyme to generate a 3.5 kb hybrid fragment. Δ*PoPKA-C* genomic DNA was digested with both *Bam*HI and *Kas*pAI enzymes, resulting in a 2.2 kb hybrid fragment. For C*PoPKA-C*, the random insertion of *PoPKA*-*C* led to the generation of 2.2 kb and 3 kb hybrid sequence fragments. (**d**) The expression of the *PoPKA-C* gene was analyzed using RT-qPCR with the internal primers RT-3213-F/R (Appendix A). T-tests were conducted to analyze differences in gene expression levels for significance (*** *p* < 0.001).

**Figure 3 jof-09-01203-f003:**
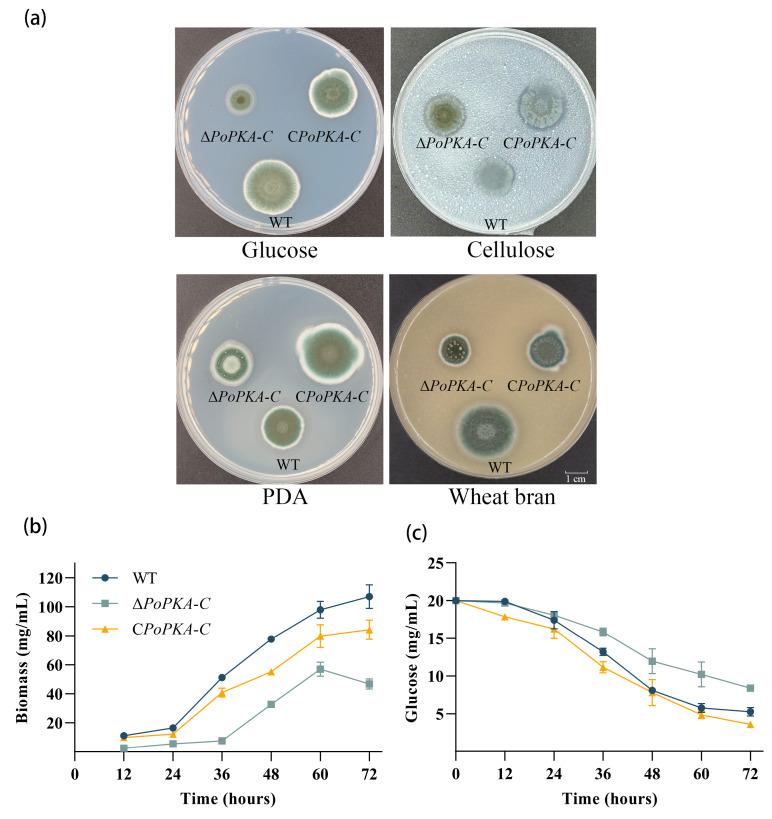
Effect of *Po*PKA-C on the growth and phenotype of *P. oxalicum*. (**a**) Phenotype of *P. oxalicum* 114-2, Δ*PoPKA-C*, and C*PoPKA-C* colonies. Phenotypic analysis was performed on MMS medium plates containing 2% glucose or 1% cellulose, PDA medium plates, and 10% wheat bran medium plates at 30 °C for 3 days. (**b**) The impact of *PoPKA-C* knockout on the biomass of the strain Δ*PoPKA-C*. (**c**) The impact of *PoPKA*-*C* knockout on glucose utilization efficiency in the Δ*PoPKA-C*.

**Figure 4 jof-09-01203-f004:**
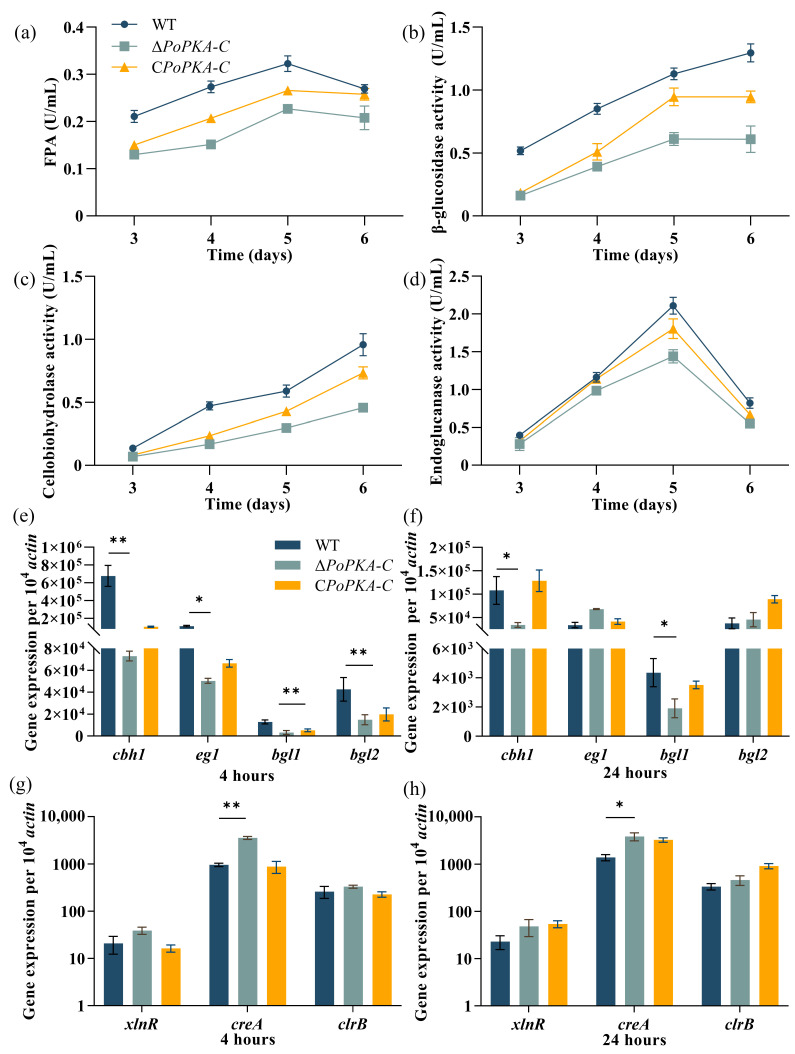
Cellulase activity analysis and expression analysis of major cellulase-related genes in *P. oxalicum* 114-2, Δ*PoPKA-C*, and C*PoPKA-C*. (**a**) FPA; (**b**) β-glucosidase activity; (**c**) Cellobiose hydrolase activity; (**d**) Endoglucanase activity; All strains were sampled and tested every 24 h from 72 h to 144 h. After 4 h and 24 h of induction with cellulose, the expression levels of four cellulase genes (**e**,**f**) and major transcriptional regulators (**g**,**h**) were measured. T-tests were conducted to analyze differences in gene expression levels for significance (* *p* < 0.05, ** *p* < 0.01).

**Figure 5 jof-09-01203-f005:**
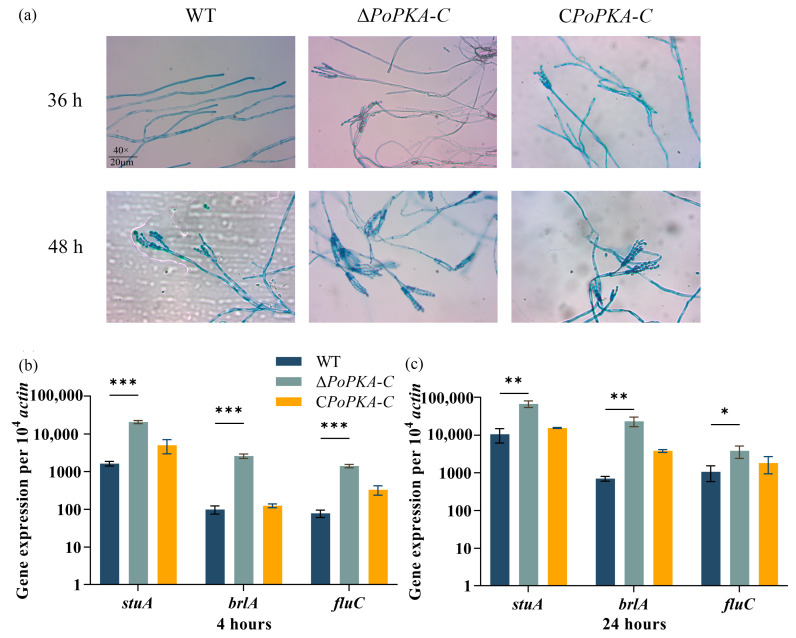
Effect of *Po*PKA-C on spore germination of *P. oxalicum.* (**a**) Microscopic observation of spore and hyphal morphology on PDA medium. After 36 h and 48 h, the microscopic morphology of spores and mycelia were observed by 400× magnification of an optical microscope. (**b**) The expression levels of *brlA*, *stuA*, and *fluC* in *P. oxalicum* 114-2, Δ*PoPKA-C*, and C*PoPKA-C* were detected by RT-qPCR, after being induced by cellulose for 4 h. (**c**) The expression levels of *brlA*, *stuA*, and *fluC* in *P. oxalicum* 114-2, Δ*PoPKA-C*, and C*PoPKA-C* were detected by RT-qPCR, after being induced by cellulose for 24 h. T-tests were conducted to analyze differences in gene expression levels for significance (* *p* < 0.05, ** *p* < 0.01, *** *p* < 0.001).

**Figure 6 jof-09-01203-f006:**
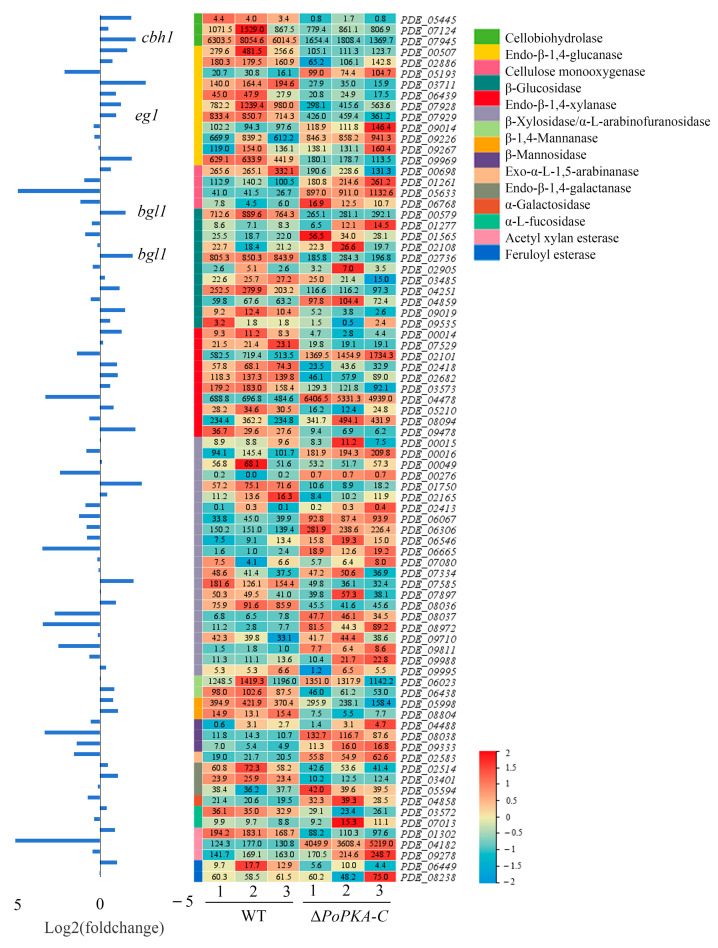
Expression of 80 annotated cellulose hydrolase genes in *P. oxalicum* 114-2 and Δ*PoPKA-C*. Levels of expression (RPKM) of the 80 cellulose hydrolase genes are indicated by a green–yellow–red color scale.

**Table 1 jof-09-01203-t001:** Screening results of MASS spectrometry.

Name of Each Protein Complexes	Annotation	ID	The Total Number of MS Profiles Matched to the Peptide
ClrB transcriptional activator	Cellulase transcriptional activator ClrB	PDE_05999	239
RNA binding protein	RNA binding effector protein Scp160	PDE_08970	136
Camp-dependent protein kinase A	cAMP-dependent protein kinase catalytic subunit	PDE_03213	121
	cAMP-dependent protein kinase regulatory subunit	PDE_04688	113
Factor of elongation	Elongation factor EF-1 α subunit	PDE_02596	95
Proteins associated with ATP	ATPase subunit α	PDE_06476	81
	ATPase subunit β	PDE_07279	77
	ATP carrier protein	PDE_02785	46
Heat shock proteins	Heat shock 70 kDa protein 1/8	PDE_02379	65
	Heat shock 70 kDa protein	PDE_03760	40
	Heat shock 70 kDa protein 12B	PDE_00429	29
Translation initiation factor	Translation initiation factor eIF-4A	PDE_01963	45
RNA polymerase II	DNA-directed RNA polymerase II subunit B	PDE_05914	43
Other	Hypothetical protein	PDE_09570	52
	Alkyl hydroperoxide reductase subunit C	PDE_05550	47
	Hypothetical protein	PDE_03634	40
	Ubiquinol-cytochrome c reductase core subunit 2	PDE_09131	36

## Data Availability

All data generated or analyzed during this study are included in this published article (and Appendix A).

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
