# Peer review of "Functional Study of cAMP-Dependent Protein Kinase A in Penicillium oxalicum"

_jof, 2023, doi:10.3390/jof9121203_

Round 1

Reviewer 1 Report

Comments and Suggestions for Authors

In this paper, the authors found cAMP-dependent protein kinase A (PoPKA-C) as a protein interacting with ClrB, a regulator of cellulase production in Penicillium oxalicum, and analyzed its function. Results show that PoPKA-C is involved in mycelial development, conidiation, and cellulase gene expression. This study seems to be important for understanding cellulase production and other physiological activity in filamentous fungi. I would like to point out the following.

1. The CPoPKA-C strain has been generated by non-homologous recombination. In this case, the position of the expression constructs, where they are incorporated into the genome, is sometimes problematic. In this case, did the authors use different independent transformants for each experimental triplicate? Also, why was homologous recombination not performed?

2. Since the expression data is normalized by actin expression, it can be regarded as expression per cell. Therefore, we can assume that PoPKA-C affects cellulase gene expression. Similarly, it would be better to show enzyme activity in terms of per-biomass. Although the medium contains insoluble cellulose, methods have been reported to measure biomass even under such conditions.

3. Although data for CPoPKA-C strains are shown in the respective figures, they are not mentioned at all in the text.

Reviewer 2 Report

Comments and Suggestions for Authors

The article is very well presented, and the research design is adequately planned. PoPKA-C deletion and complementation strains have been used to elucidate the role of phosphorylation in the regulation of lignocellulases synthesis.

Interestingly, the use of affinity chromatography has revealed ClrB complex proteins. However, the results have not been discussed, beyond the interaction with PoPKA. Are there previous reports of functional interaction of ClrB with the other proteins reported in Table 1? Please include those results in the discussion.

M&M

Please indicate the time of incubation for the Penicillium oxalicum culture in cellulase-induced fermentation medium (section 2.1).

Conclusions should be included.

Round 2

Reviewer 1 Report

Comments and Suggestions for Authors

The authors responded appropriately to the reviewers' remarks and made the manuscript better. Hence, this manuscript is acceptable.